# Autonomy, Motivation, and Digital Pedagogy Are Key Factors in the Perceptions of Spanish Higher-Education Students toward Online Learning during the COVID-19 Pandemic

**DOI:** 10.3390/ijerph19020654

**Published:** 2022-01-07

**Authors:** María Dolores Díaz-Noguera, Carlos Hervás-Gómez, Ana María De la Calle-Cabrera, Eloy López-Meneses

**Affiliations:** 1Department of Didactics and School Organization, Faculty of Education, University of Seville, 41013 Seville, Spain; noguera@us.es; 2Department of Language and Literature Teaching, Faculty of Education, University of Seville, 41013 Seville, Spain; anamariadelacalle@us.es; 3Department of Education and Social Psychology, Area of School Organization and Didactics, Pablo de Olavide University, 41013 Seville, Spain; elopmen@upo.es

**Keywords:** active pedagogies, motivational factor, higher education, TIC, empowerment, teaching competences

## Abstract

This paper proposes a development model of the adaptation capacity of students to digital transformation in university teaching through three constructs: motivations, digital pedagogy, and student autonomy. For this study, an ad hoc scale was created to record the adaptation capacity of students to digital transformation. The sample was 483 students from the University of Seville (Spain), to whom an online survey was administered during the development of online teaching in the period of November 2020 using the Google Forms platform. The findings of this study showed that university student motivation acquired a greater threshold than autonomy, whose threshold in turn, was greater than that of digital pedagogy in the ability to adapt to online teaching and that the capacity of adaptation to the online modality is explained by the perception that university students have of the usefulness, products, and learning outcomes, among others. In conclusion, the lack of adequate and enabled study spaces is key to developing the online model. We consider all these aspects as prospective research objectives.

## 1. Introduction

At the height of the COVID-19 pandemic, university students and faculty were forced to learn and teach online. Regarding the adaptation of the educational community, our questions are focused on students. What factors favor the adaptation capacity of students to the new higher education model? What is the relationship between self-regulated learning and autonomy in university students? Is it necessary to develop a pedagogical design as the core idea, where challenges can be overcome such as the detachment of online tasks from face-to-face tasks? In this sense, the following specific objectives are set to define a model of the adaptation capacity of students toward digital transformation in university teaching.

The background for this type of question is found in systematic reviews on these topics [1]. E-learning had already contributed to virtual environments with its strengths and limitations [2]. One of the most important advances was the development of communication and interaction tools [3], which allowed access to teaching resources that have enriched university learning and the multidisciplinary and global understanding of knowledge. However, after one year of pandemic and experiencing the multimodal or hybrid paradigm, it is necessary to further investigate a series of elements that are essential to achieve quality teaching in higher education.

Previous works [4] had the purpose of studying the perceptions of students in relation to the educational change that they had suffered abruptly due to the scourge of COVID-19. The systematic reviews that have been carried out [5] coincide with our research at the point of analysis on how students had to adapt to this new educational situation, how not all educational models were coherent, and the results of the investigations. The key elements of the hybrid educational models were questioned, pointing out the importance of integrated design, an adequate virtual environment, the management of physical spaces, tutoring, and continuous evaluation. The educational digital transformation has revolutionized traditional forms of teaching, incorporating new questions into educational research and opening new spaces. It marked a turning point in the global educational community, according to research on emergency remote teaching (ERT). The topics addressed pointed to new learning designs, the institutional culture, evaluation systems, professional collaborations, and student empowerment.

First, we want to highlight that to enable students to be empowered, it is necessary to acquire digital competence at an instrumental level, as indicated by previous studies in which a sociocultural model is designed for the development of digital competence [6]. The first implication of this model in relation to our proposal is that, to guarantee the development of digital competence at an instrumental level, university teachers must include activities that allow students to handle digital tools at a basic and technical level. This is what we call student autonomy in the model we plan.

Second, at a strategic and operational level linked to the activation of students in digital skills, it is necessary to use digital resources to the point where students can say they are in the so-called comfort zone. Reviewing classical authors such as [6], we find that the subjective and implicit processes of decision-making by students highlight the degree of internalization of digital skills. The second implication is the generation of scenarios in which the application of these tools is required until the students activate the choice spontaneously or forcedly, as has been the case with the COVID-19 pandemic. In any case, for the students to be empowered, they must be autonomous, as we have indicated, and they must have mastery and digital appropriation. Additionally, the process will be successful if the appropriate spaces are generated (active dynamics), which provoke the second dimensions of our model, namely the motivation of the students, since we have caused the choice and therefore, the identification of preference, adding the reinterpretation of learning objects that are capable of designing and being applied in their networks or the close environment causing the social impact.

Students have been able to expand their individual and social capital through technology, except for the existing gap in students at risk of exclusion that research has revealed. In the literature review, we have been able to verify that the exchange and internalization process increases when students have digital skills [6]. At this level, subjects have appropriated digital culture by assuming its rules, identifying its characteristics, and being part of them. They construct their own meaning and internalize their narratives that they are autonomous. These relationships are reflected in Figure 1.

In this sense, redefining the times in the execution of tasks by students becomes essential, as shown in previous studies conducted in Canada [1], where this aspect was highlighted as a major limitation in achieving genuine learning. The challenge is to achieve maximum efficacy in learning. The fourth element is autonomous workspaces for students. These autonomous working spaces or digital learning classrooms will be equipped with autonomous work guidelines, didactic materials with a broad audiovisual culture, and communication resources, all enhanced by the transformation of the evaluation process and the support of good tutorship.

Specifically, we developed the following three sections: the motivation of future teachers, a review of the current state of digital pedagogy, the elements we consider to be key for quality online learning, and the competences that are being incorporated into the future teacher profile.

Digital transformation (DT) could be defined as the set of actions that lead society toward the new era. Everyone is immersed in this new milestone, which is drastically changing society; therefore, education is not exempt from these modifications. Educational digital transformation (EDT) is based on remote work, teleworking, and online learning, i.e., professionals of the 21st century. Therefore, the aim of the present study was to know and interpret the behaviors and feelings of future teachers when dealing with multimodal or hybrid models since our purpose is to continually develop the best educational practices [7].

This implies becoming proactive in their work life, responsible in their own self-development, and self-managers of their intentional learning projects inside and outside formal institutional environments. According to this description, the aim of a number of the international scientific journal is “to determine the conditions that promote the implementation and results of innovation processes, based on their diverse dimensions”. We are still facing the challenges that have led researchers to pursue deep learning for the last two decades. University students will achieve deep and time-stable learning if universities promote the collaborative construction of learning, integrating the previous knowledge and establishing the learning experiences. Attaining this goal requires, first, that faculty members appreciate the nature of this conception of learning, and second and most importantly, that they plan their teaching while considering this transformation.

### 1.1. Motivations

Decades since, authors like [8] have contributed to understanding the relationship between motivation and self-regulated learning (SRL). Self-regulation of learning is not a mental capacity such as intelligence or a skill such as reading but a self-guided process through which learners transform their mental capacities into academic skills. We reviewed the general framework to dive into the relationship between motivation and SRL. According to this framework, SRL facilitates the achievement of the educational goals set by the different disciplines’ programs from an intrinsic orientation, that is, driven by the student’s own motivation. Similarly, SRL has an extrinsic character as well due to the value granted to the assignments. These can be connected to the pedagogical design and integrated into the educational process, where they will be meaningful to the student and will empower the latter. Therefore, this is called “locus of control for learning” which consists of achieving maximum effectiveness and performance in students and incorporates a detailed analysis of their emotions to obtain the desired marks. SRL is defined as the set of strategies that students use to regulate their cognition (that is, the use of several cognitive and metacognitive strategies) as well as the set of resource management strategies that students use to control their learning.

Academic self-regulation has been established as a basic competence in the education system under the paradigm of the European Higher Education Area (EHEA) for over twenty years and is fundamental today due to the special circumstances of the pandemic. Therefore, it is important to determine which factors should be taken into account in SRL studies and what other concepts would be related.

In this sense, SRL involves student autonomy, especially competencies such as thinking, cooperation, communication, empathy, being critical, and self-motivation. The latter, as an attribute of the competence of “learning to learn”, is directly related to the term “self-regulated learning” (SRL), which was coined from the social-cognitive approach to learning to refer to the active and constructive process in which students set their own learning goals and then try to monitor, regulate, and control their own cognition, motivation, and behavior, guided by their objectives and the characteristics of the environment [8]. Moreover, some authors define successful university students as “academically self-regulated students” [9]. Our actions can be triggered by intentions, rewards, or intrinsic values, as stated by [10], which highlight the need to include this knowledge in the practices we carry out in higher education.

Recent neuroscientific studies have shown some results about growth mentality and intrinsic motivation. With the advancements in neuroscience and motivational studies, there is a global need to use this information in educational practice and research. However, little is known about the neuroscientific interaction between growth mentality and intrinsic motivation. SRL is also associated with creativity, especially with organizations, as shown by [11], whose results confirm the positive impact on the effects of intrinsic motivation and creative and motivational performance. In universities, the current pandemic has forced the accelerated incorporation of different digital models (hybrid models, i.e., sharing physical spaces and virtual working environments). What caused this situation in the development of digital competencies of university students? Fundamentally, students have developed competencies in the self-management of innovating and emerging digital methodologies such as the TPACK theory (Technological Pedagogical Content Knowledge) and the SMART goal (Specific, Measurable, Achievable, Realistic, and Timely). Multiple studies have concluded that the use of these methodologies changes the thinking models of university students, integrating autonomous learning and new technologies [12].

There are correlations between the autonomous educational processes of university students and creativity. These assertions are shown by studies conducted on student motivation and evaluation. Thus, students understand evaluations as a process of personal development and are motivated by the achievement of a creative learning goal [13].

We have revised and verified the impact of learning styles on future leadership models. The attitudes toward computer technology are key as well as the design and creation of materials. It is important to take into account the relationship between interests and skills in order to choose the most adequate education and evaluate its strategies [9].

It becomes clear that neuroscience plays a relevant role in education. Educational neuroscience helps to understand how the brain works and how learning is influenced by neurobiological processes. Therefore, it is another concept that is incorporated in research on the motivation of university students.

The literature on advances in neuroscience and motivation shows that different researchers demand the inclusion of this knowledge in professional educational practices. For example, student performance has been stated to be higher in centers that promote positive leadership and a mentality of professional growth and development through the design and implementation of change and improvement projects [13].

Digital citizenship, i.e., the competencies and ethical values required to participate in an online society, is an increasingly essential element in the 21st century. Critical thinking [14], citizenry [15], and the inclusion of systems such as interactive groups, collaborative learning, and peer tutoring have proven to be efficient strategies that help all students achieve their maximum potential based on their learning capacities, while they also promote social inclusion and the coexistence of the entire classroom and community [16].

### 1.2. Active Digital Pedagogies and Student Autonomy

Active pedagogies are a field of transformation and change in education. Considering how organizations adapt to the new digital era, active pedagogies constitute an experimental field. Researchers concerned about the educational scope have focused their studies on reviewing the term “digital learning” [17] and the incorporation of collaborative learnings [18]. The perceptions of students and educators toward digital learning have been a key aspect in the research hypotheses of the last decades, and they currently have profound implications for digitalization in education.

Personal learning environments (PLEs) have become essential experiences during the pandemic. In this context, studies are aimed at critically analyzing the new learning environments as dynamic and informal environments immersed in the learning ecologies (LE). Therefore, rethinking the curriculum and LE as an analytical framework to determine how we learn and what contexts we need is a relevant strategy for digital learning [19,20].

Regarding interactive learning environments, different investigations have focused on the design of electronic learning activities (E-shop), following Piaget’s cognitive theories and Vygotsky’s social constructivism. Other authors have focused on flipped classrooms or flipped, deep, and improved learning with mobile technology and the combination of active pedagogies [3]. However, relationships with communities have been one of the most important concerns [21]. Communities are made up of users who share similar visions and behaviors and form knowledge networks. Thus, changes in interactive learning environments are ecologically related between individuals and communities. In this sense, studies have been focused on academic performance, the efficacy of learning (both cognitive and emotional), satisfaction, and self-efficacy. Baturay [21] identified a strong relationship between the proposed content and the students in interactive learning. Consequently, it is especially interesting to find out how learning ecologies and the communities that comprise them are adaptive and adjust to changes in the teaching environment brought about by COVID-19, particularly in higher education and the training of future teachers.

## 2. Materials and Methods

### 2.1. Method

This is a nonexperimental, descriptive, survey-based study [22]. The objective of this research is to verify the relationship between the factors, motivation, autonomy, and digital pedagogy in the adaptation of university students to online education in the state of emergency that arose as a result of the COVID-19 pandemic. To this end, an online survey was administered to university students during the development of online teaching in the period of November 2020 using the Google Forms platform. The sample was recruited by nonrandom selection. Specifically, a nonprobabilistic, causal sampling was performed in which the most common selection criteria were based on the accessibility of the participants. In particular, the participants of this study were 483 students from the University of Seville (Spain), with 348 women (72%) and 135 men (28%) aged between 18 and 25 years (mean = 20.7 years), who were registered in social and health sciences degrees in the academic years of 2019–2020.

In this academic year, the number of students enrolled in the University of Seville reached 70,900. Thus, considering a heterogeneity of 50%, a margin of error of 5%, and a confidence level of 95%, the selected sample is representative of the said population since it is greater than the 383 necessary cases.

### 2.2. Instrument

To collect data that respond to the proposed objectives, we used an ad hoc scale that allowed us to evaluate the perception of university students toward online teaching in COVID-19 scenarios and which was used in our previous research [4] and was based on the literary framework itself and previous research [23,24]. In particular, we were inspired by the design of the items on the scale designed by [25] for the analysis of digital transformation and the Motivated Strategies for Learning Questionnaire of [24] to value the motivation component toward online teaching. This instrument consisted of 37 items (5 identification items and 32 items about digital transformation), grouped into five categories: student profiles, resources (hardware–software), professional collaboration, digital pedagogy, and student empowerment (motivation). Regarding the student profiles, five questions were included to gather information about the main characteristics: (1) sex, (2) age, (3) degree year, (4) group, and (5) degree. The students answered the rest of the questions in a Likert-scale from one (strongly disagree/little) to five (strongly agree/much). The new items were recoded before conducting the analyses (items 9, 21, 22, and 28). The Cronbach’s Alpha obtained was 0.73. To respond to the objectives of the study, the instrument was used for the analysis of the three constructs proposed in the theoretical section: motivations (student empowerment), digital pedagogy, and student autonomy (professional collaboration). Thus, initially 27 items were considered in three constructs for the delimitation of the model (see data analysis section). However, the items in the three categories were organized in the model confirmation process as follows:(1)Motivations (student empowerment)

Item 1. It is important for me to learn the topics of this subject.

Item 2. I am very interested in the contents that I am learning in this subject.

Item 3. I am sure that I can do a great job in the assignments and exams of this subject.

Item 4. I think that the material of this subject is useful to learn.

(2)Student autonomy (professional collaboration)

Item 5. How does online education affect your interactions with your classmates? (extremely bad–extremely well).

Item 6. In my opinion, I learn better in face-to-face lectures than in online lectures.

Item 7. How important is it for you to interact with the teacher verbally every week?

(3)Digital pedagogy

Item 8. How difficult is it to adapt to online practical work/activities/assignments?

Item 9. Do you think that online education is useful?

Item 10. How much are you enjoying online education?

The three factors define a model that determines the capacity of the students to adapt to digital transformation in university teaching through their perceptions (Figure 2). Thus, our scale reports on the adaptation capacity of students toward digital transformation, considering the level of digital competence that university students show through their preferences and capacity to adapt to the digital world, their motivations, and expectations toward learning the content, and their capacity to acquire an autonomous role in the model of online teaching.

The resulting model derives from the solution obtained in the CFA based on the first examination (CFA), undergoing adjustments in the included items (based on their factorial load) and their organization within the factors (based on the theoretical dimensions), as indicated in the next section on data analysis. Thus, the model derived from this first analysis was tested, which is made up of three factors. Of the 17 items that were integrated into the three factors, ultimately 10 of these were included in the confirmed model since a satisfactory adjustment solution was obtained. Likewise, a unidimensional model was estimated that presented worse fit indices χ^2^ = 1470.675, *p* < 0.001, root mean square error of approximation (RMSEA) = 0.164 (95% CI: [0.15, 0.18]), comparative fit index (CFI) = 0.60, goodness of fit index (GFI) = 0.77, Bentler-Bonett non-normed fit index (NNFI) = 0.50, and standardized root mean square residual (SRMR) = 0.15. Consequently, based on the results obtained, the model of three correlated factors, made up of 10 items, was chosen to show better fit indices as detailed below.

Firstly, the confirmed model presents a Chi-squared distribution with values of χ^2^ = 1338.776 and *p*-value < 0.001. The CFI, TLI, and NNFI of the model obtained the following values: CFI = 0.94, TLI = 0.91, and NNFI = 0.91. Thus, the explanatory model obtained good parameters of fit in these indices. The RMSEA obtained a value of 0.07 (95% CI: [0.05, 0.08]). The values of GFI and SRMR was 0.96 and 0.06. Both indicate good results of goodness of fit as well.

### 2.3. Data Analysis

The data were subjected to path analysis and descriptive analyses (frequencies, mean, maximum and minimum, standard deviation, and variance) using SPSS version 26 (IBM, Chicago, IL, USA) and JASP software version 0.11.1.0. JASP (JASP, Amsterdam, The Netherlands) was used to define an explanatory model about the study object, and SPSS was used for the descriptive statistical analyses of the mean, variance, standard deviation, range, and minimum and maximum values.

At the procedural level, firstly, a confirmatory factor analysis (CFA) was conducted making use of all the items on the scale. Thus, this analysis considered three factors for the organization of 27 items, with inadequate fit indices (RMSEA = 0.103 and TLI = 0.55). Consequently, with these results, an exploitative factor analysis (EFA) was carried out to determine if there was a possibility that the items were being organized in another structure. To estimate the exploratory model, three factors were defined manually, following the minimum residual estimation method and promax oblique rotation and obtaining the structure of the scale in essential but reducing the items that contributed weight to the factors. Thus, this analysis helped to propose a reference structure with a smaller number of items. Also, items with factorial loads less than 0.3 were excluded from the model.

Then a new CFA is performed based on the EFA until the best goodness-of-fit indicators are achieved, which will reduce the initial theoretical design proposal from 27 items to 10. Thus, it was revealed that the model of adaptability of university students to online teaching was made up of three factors explained by 11 items on the scale (for more details: go back to the instruments section). To determine the goodness of fit of this model, we analyzed additional fit parameters, the goodness-of-fit index (GFI), and the coefficient of determination (R^2^). We took as reference the criterion established by [26,27]), according to which, good values of fit are obtained if GFI, CFI, TLI, and NNFI ≥ 0.90 or 0.95, and RMSEA ≤ 0.05 to 0.08. Regarding the standardized root mean square residual (SRMR), we applied the criterion of Hair et al.) [28], who consider that values equal to or lower than 0.08 indicate a good fit.

## 3. Results

Firstly, in the previous section, where the data collection instrument is presented, we respond to the first objective of the work: to formulate a scale to discover the adaptation capacity of students toward digital transformation in university teaching based on a theoretical model that integrates the constructs: motivations, digital pedagogy, and student autonomy (Figure 2).

Regarding the second objective about determining the valuation of students’ digital transformation that occurred in university teaching as a result of COVID-19, the descriptive results incorporated in Table 1 reveal that most of the average scores of motivations are around 4 (above the mean value of the scale), the scores of autonomy are more dispersed (around 3 in the middle), and the scores of digital pedagogy are around 2.5. The different factors of student performance showed a tendency where university student motivations acquired a greater threshold than autonomy, whose threshold, in turn, was generally greater than that of digital pedagogy. Similarly, it is worth highlighting that the perception of students toward motivation for the subject matter is above their self-perception toward their thresholds of autonomy and digital mastery in this context of unpredicted online modality. Within the motivation constructor, the items, “I am very interested in the contents that I am learning in this subject.” (average = 4.34) and “I am very interested in the contents that I am learning in this subject.” (average = 4.34), obtain the highest average scores, which values the motivation of the students toward the knowledge of the subjects. However, despite this reason, the development of good products in them and the learning functionality cease to be an incentive since as can be seen in the items “I am sure that I can do a great job in the assignments and exams of this subject.” and “I think that the material of this subject is useful to learn.” scores are also higher than four points.

On the other hand, in the autonomy construct, the item “How does online education affect your interaction with your classmates?” obtains the lowest average score, encouraging the perception of university students of the limitation that the online modality supposes for the interaction between equals. Also, the item “In my opinion, I learn better in face-to-face lectures than in online lectures” (average = 4.41), warns the students’ perception of the limitation that this modality has for effective learning. On the other hand, in the same construct, the item “How important is it for you to interact with the teacher verbally every week?” (average = 4.47) manifests the students’ perceived need for a fluid relationship with the teaching staff that can compensate for the limitations found in the digital transformation.

Finally, in relation to the digital pedagogy construct, all the items are at the medium–low threshold. In particular, the item “How difficult is it to adapt to the situation of online theory lectures?” (average = 2.29), reports the students’ perception of a good ability to adapt to online teaching when what is carried out are online theoretical lectures, while the items “Do you think that online education is useful?” (average = 2.85) and “How much are you enjoying online education?” (mean = 2.38) show the disconnection of students toward the usefulness and enjoyment of online teaching.

In relation to motivations, Figure 3 shows that around 70–80% of the students agree completely that the material of the subjects in this modality is functional, that the contents are interesting, that it is important to learn about them, that they will do well on papers and tests, and that they are concerned about their grades. This indicates that the ability to adapt to the online modality is explained by the perception that they have of the usefulness of the products and learning results being the motor of this adaptation to the latent concern in the students for overcoming the subject in the online modality.

At the level of student autonomy, more than 60% of students reveal preferences for this format and consider it as promoting interactions between students in class sessions while at the same time consider that contact with the teacher is not relevant in the same percentage (Figure 4). This shows that students develop an autonomous role vis-à-vis the teacher as well as greater collaboration networks among peers to achieve the shared objective: learning the subjects in the online mode. Thus, the degree of adaptation of university students to online teaching depends on their perception of their self-sufficiency and collaborative work in learning the subjects.

Finally, at the level of digital pedagogy, more than 60% of students consider online education useful. However, around 50% of the students find limitations in terms of adaptation and enjoyment of this teaching modality (Figure 5). These results could indicate a recognition by students of the usefulness of an active digital pedagogy that implies an interactive methodology in the use of digital resources, but they could also indicate that students do not want to give up the face-to-face teaching modality even though they find the online modality functional. This highlights the fact that the lesser or greater adaptation of the students to this modality could reside in the dichotomy of cost or enjoyment in the digital transformation process.

## 4. Discussion

This research is caused by the COVID-19 pandemic and looks to contribute to the studies carried out on this subject at the university level [5]. In a previous article [4] we identified the perceptions of university students toward teaching–learning processes, influenced by the changes suffered in the hybrid education model to which they were subjected during the state of alarm. The results obtained led us to analyze the factors presented in this article. Thus, the factors that favor the adaptation of students to digital transformation were professional collaboration, digital pedagogy, student empowerment, self-learning, and the promotion of initiatives that promote the development of future teachers.

On this occasion, we present a model that relates three factors for digital adaptation in university teaching which are: autonomy, professional collaboration, motivation, and active pedagogies. This model determines the ability of students to adapt to digital transformation; it is necessary to take into account the level of digital competence shown by university students through their preferences and capacities to adapt to the digital world, their motivations and expectations toward learning, and their ability to acquire an autonomous role in the online teaching model. We have also seen how online education affects peer interactions. Peer interactions should be favored in situations where personal contact was not possible. The development of active pedagogies is the challenge that we must overcome. The results have indicated or called our attention to how not being present has had an impact on the effective learning of students. The students felt the need to interact verbally with the teachers at least once a week; undoubtedly, this gives us clues about the need to improve the online pedagogical model.

We had experience in the hybrid multimodal method. Some of their results indicate that in these educational models, different learning strategies are offered to train students in competencies such as the search for relevant information, cooperative work, decision-making, and elaboration of content [29]. However, in our data we do not find these benefits; in the same way the description of favorable contexts or environments does not appear in the data where the needs of the students have been channeled. We have identified that the methodologies used have allowed for synchronous teaching where face-to-face classes and asynchronous training are being developed in virtual environments and where students can interact with each other, access the content of virtual teaching, and perform the tasks. Finally, two of the greatest challenges posed by online education are the digital divide and the lack of support from the institutions, something that greatly conditions the success of the teaching process since, if the institutions do not promote an update of the teachers’ knowledge regarding technological tools aimed at creating educational materials suitable for virtual environments, we will be facing the aforementioned case in which online classes are a virtual imitation of face-to-face classes [30].

This work presents us with the problems and solutions offered in research on the COVID-19 pandemic [31]. The use of technological tools was key to ensure that the education sector did not suffer significantly from the mobility restrictions imposed by the Spanish government during the COVID-19 pandemic. This led to effective virtual teaching and the use of tools that many teachers had never thought of incorporating into their teaching and training and which could continue to be used after returning to traditional training [32].

In general, the participants of this study consider that they will obtain good results with online teaching and are highly motivated with their studies. However, some of them find it difficult to adapt, feel discouraged (they do not like it), and reject this way of learning for the future. Moreover, they highly value the interactions with their classmates and teachers in the classroom, as well as physically being in the classroom, which is in agreement with the results of other studies conducted throughout the world [33]. Our goal is to continue delving into the relationship between self-regulated learning, intrinsic student motivation, and self-evaluation.

## 5. Conclusions

We should not forget the importance of knowing the interests of students in DT in certain educational areas. This requires the conceptual and philosophical re-evaluation of teaching and learning as well as of the roles of teachers, students, and didactic materials and the connections between them [34]. Some recommendations are given to organizations: good communication, providing information about the change, involving students in making decisions related to the transformations carried out, adjusting the content and teaching method to the way of online learning, taking care of social presence using synchronous forms, limiting the tools used (preferably to choose one), and providing support in the field of technologies used which enables participation in online learning [35].

In this sense, future research should further explore the design of autonomous work guidelines for higher education students and advance didactic resources, digital pedagogies, and fundamentally the concept of evaluation, as is discussed in this article. We conclude that factors, such as university student autonomy, are the ones that improve learning performance the most, and they involve a certain level of adaptability to the new requirements of educational digital transformation. Moreover, due to the importance of the learning design, the emerging and innovative digital pedagogy is a phenomenon that promotes our incorporation as professionals of the 21st century to the new educational era. The results of this study suggest a set of priority areas that require attention in order to improve student satisfaction with online training [36].

It is necessary to dive into the characteristics of what many authors have defined as “digital learning” [17] and collaborative learning [37], including their role-play design, which should be included in future studies. Similarly, the exchange and relationships that take place in joint learning [18] include suggestions for community management learning. The proposed selection of topics paves the road for further work on future teacher training, the evaluation of our students, and how to continue incorporating collaborative tools and improve our digital learning designs with other social networks. In this sense, it is important to continue collaborating with the creation of learning communities. These relationships with the community are very relevant due to the types of interactive learning [21]. In this sense, studies have focused on academic performance, the efficacy of learning (both cognitive and emotional), satisfaction, and self-efficacy. Baturay [21] identified a strong relationship between the proposed content and the students. Other studies point out the difficulties of performing collaborative group assignments; in many cases, evaluations and the search for better individual results hinder collective intelligence management projects.

The implications for educational institutions is to require a good integrated design of the teaching process as well as requesting the administration to provide the centers with the necessary digital resources to achieve digital transformation as soon as possible. The most repeated sentence through the pandemic, “leave nobody behind”, must become reality. It is fundamental to create autonomous workplaces or digital learning classrooms that allow the incorporation of quality education into the new era.

Higher education cannot forget that the impact of Industry 4.0, characterized by the digitalization of cybernetic processes and systems, is based on three trends: artificial intelligence, immersive transparent experiences, and digital platforms. It is necessary to promote motivation and autonomy in the learning processes of future educators. The fourth industrial revolution is also a challenge, as it requires the development of disciplinary and transversal competencies that can only be acquired in learning processes throughout their life, constantly preparing them for a changing and demanding work reality, a true EDT in all teaching–learning processes in higher education, as we identified in the literature review [38,39]. Contrary to what one might think, the EDT is a pending account in higher education. Online examinations, commonly referred to as e-exams (electronic examinations), underwent a considerable progression, being adapted ubiquitously among higher education institutions worldwide. Their preference was rapid due to the emergence of the COVID-19 pandemic. The online examination process is being adopted as the appropriate way of evaluating, while ensuring the safety and well-being of students [40].

In reviews of research carried out on this topic, we have found that there are greater difficulties when it comes to achieving the best results from students. What factors have made it difficult to get the best results? These factors include ignorance of the educational model and the methodology to be used, loss of face-to-face social contact among students, and the absence of adequate and enabled spaces for study. We consider all these aspects as prospective research objectives to find the best measures to develop to inspire administrative and educational institutions.

## Figures and Tables

**Figure 1 ijerph-19-00654-f001:**
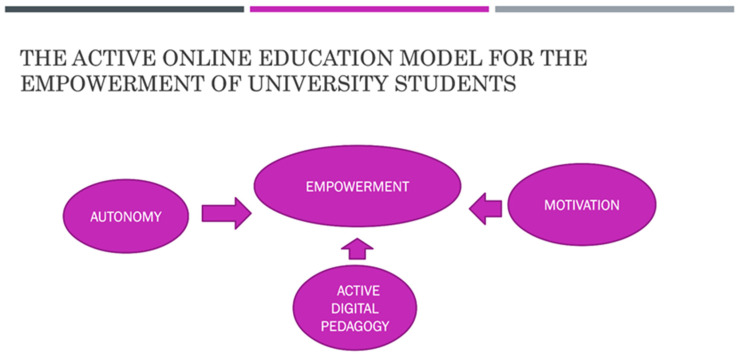
The active online education model for the education of students.

**Figure 2 ijerph-19-00654-f002:**
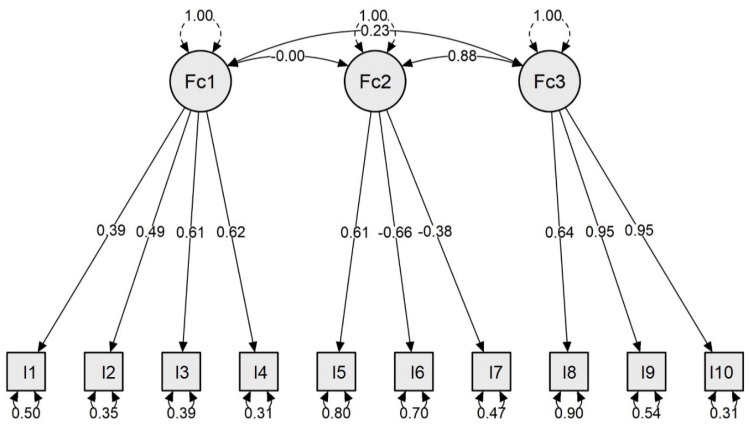
Model of the capacity of students to adapt to digital transformation. Fc1: Factor 1 or motivations; Fc2: Factor 2 or student autonomy; Fc3: Factor 3 or digital pedagogy.

**Figure 3 ijerph-19-00654-f003:**
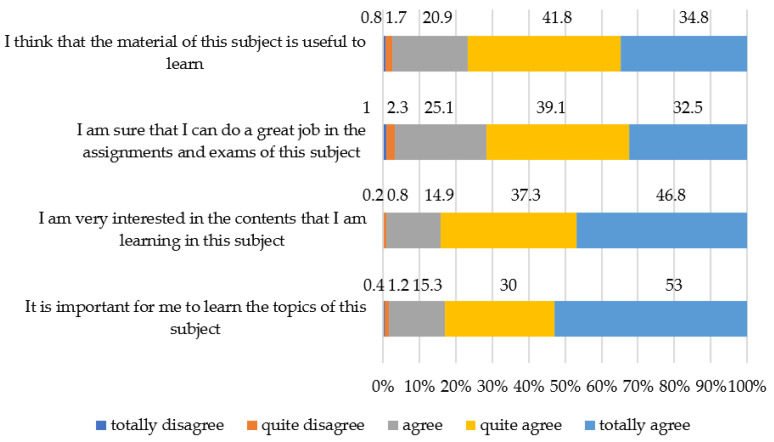
Level of ability to adapt to online teaching linked to the motivations construct.

**Figure 4 ijerph-19-00654-f004:**
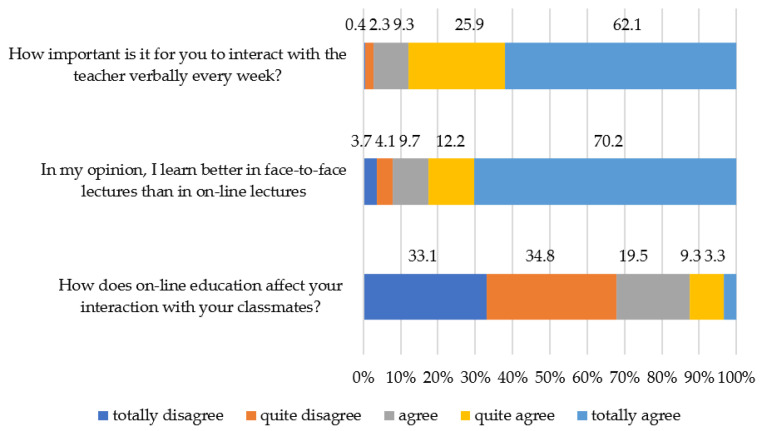
Level of ability to adapt to online teaching linked to the construct of student autonomy.

**Figure 5 ijerph-19-00654-f005:**
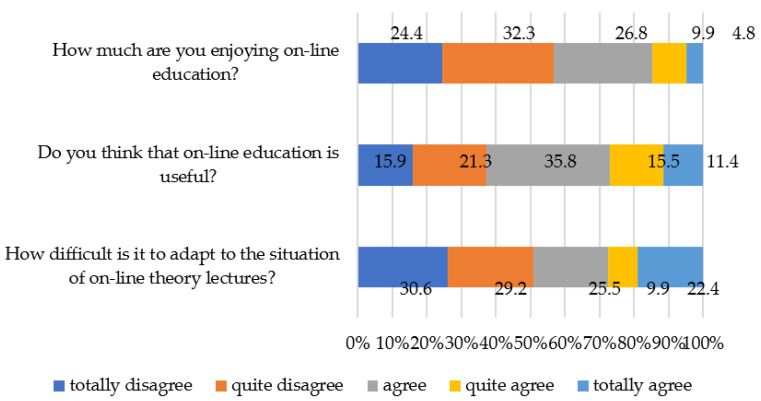
Level of ability to adapt to online teaching linked to the construct of digital pedagogies.

**Table 1 ijerph-19-00654-t001:** Descriptive data.

Construct	Item	Min.	Max.	Mean	SD	Variance
Motivations	It is important for me to learn the topics of this subject.	1	5	4.34	0.81	0.66
	I am very interested in the contents that I am learning in this subject.	1	5	4.30	0.76	0.58
	I am sure that I can do a great job in the assignments and exams of this subject.	1	5	4.00	0.87	0.76
	I think that the material of this subject is useful to learn.	1	5	4.08	0.83	0.69
	Total	1	5	4.17	0.61	0.37
Student Autonomy	How does online education affect your interactions with your classmates? (extremely bad–extremely well).	1	5	2.15	1.09	1.18
	In my opinion, I learn better in face-to-face lectures than in online lectures.	1	5	4.41	1.07	1.14
	How important is it for you to interact with the teacher verbally every week?	1	5	4.47	0.79	0.62
	Total	1	5	3.68	0.52	0.27
Digital Pedagogy	How difficult is it to adapt to the situation of online theory lectures?	1	5	2.29	1.14	1.31
	Do you think that online education is useful?	1	5	2.85	1.20	1.44
	How much are you enjoying online education?	1	5	2.38	1.10	1.21
	Total	1	5	2.51	0.95	0.90

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
