# Peer review of "Autonomy, Motivation, and Digital Pedagogy Are Key Factors in the Perceptions of Spanish Higher-Education Students toward Online Learning during the COVID-19 Pandemic"

_ijerph, 2022, doi:10.3390/ijerph19020654_

Round 1
Reviewer 1 Report
Thanks for inviting me to review this manuscript. This paper aims to investigate the acceptance of online teaching during the Covid-19 pandemic in a particular Spanish university. It is an interesting topic but there are quite a few issues that must be addressed in the current manuscript. Please see my detailed comments below.
Major comments:
- In the introduction section. The authors have made too much effort on showing the importance of online teaching, which I think can be very well explained in one short paragraph. Whilst the much more important task for the introduction section (especially when you don’t have a literature review section) is to build the theoretical links between your key concepts: motivations, digital pedagogy, and autonomy. However, although you introduced each one of them, their interactions are at least not clearly stated. That is a prerequisite for your audience to understand your work.
- It’s rude to say but unfortunately, I don’t think the model you proposed can answer your research questions. If the purpose is to verify the relationship between the three influencing factors and the acceptance of online teaching, the authors should at least have another construct “Acceptance” or “Adaption” with 3-5 corresponding items. Also, looking at Figure 2, I wonder if the authors have done a proper exploratory analysis prior to the structural model. Because if you did, items with an extremely low factor loading (such as 1, 2, 3, and 11) should be eliminated because they are not good indicators for the corresponding constructs. Therefore, I suggest the authors follow a normal SEM procedure to conduct the analysis instead of expecting an effective shortcut.
- I actually don’t understand why descriptive results can appear after the modelling results. If the authors consider the descriptive results as their main findings I must say, descriptions of some 500 students’ answers to 13 questions should never be publishable work. Please do some analysis instead of typing in their answers and describe it a little bit.
- I can only give comments on the discussion and conclusions section when the authors redo the model and conduct a proper analysis. The authors seem to be quite creative in discussing and concluding things. However, it can only be a good paper if these discussions and conclusions are based on robust analyses.
Minor comments:
- Figure 1: Why should I know these? It can be useful if you provide information about whether they are from a richer family or a poorer one. Or their household size?
- The English throughout the paper is a little bit weird. Editing service may be needed.
Reviewer 2 Report
MDPI_ijerph-1506310-peer-review-v1 Review 1 Comments
It is difficult to stop the pandemic in the short term, The importance of online teaching is self-evident. Teachers and students are more familiar with the online education model and have gained more experience now, but there are still a lot of issues that need improvement. So, the significance of this study is very important. This study proposed a model of the adaptation capacity of students toward digital transformation in university teaching through three constructs (Motivations, Digital Pedagogy and Student Autonomy). However, there are some problems and suggestions as follows:
- Problems:
- There were many clues mentioned in the introduction that supported the significance of this study. However, some clues do not seem to be directly related, such as industry 4.0, which more referred to IoT or smart manufacturing. In addition, the reasons focusing on the three constructs of Motivations, Digital Pedagogy and Student Autonomy, seemed to lack deductive logic based on literature review.
- The discussion content in “1.1. Motivations” is less relevant to the questions of items 1-5. For example, which item is related to the content mentioned on line 176, “they are motivated by the attainment of a creative learning goal.” Even the items were inspired from the reference [25],the reasons for choosing the five should be explained. Similar problems also appear in the items of digital pedagogy and student autonomy.
- How to obtain the results with significant differences in the scores of the three constructs? Maybe one-way ANOVA and post-hoc analyses would be used to get Motivations > Student Autonomy > Digital Pedagogy.
- In the discussion, the result of the questionnaire should be analyzed and then put forward some suggestions for online learning. The contribution of this study is not very clear.
- Suggestions:
- The subjects of the study came from nine different fields. Although the researchers included them in the field of social and health science degrees, there are still significant differences in these nine areas. So why did the researchers choose subjects from these nine areas? Or, what is the reason for this division? In addition, are there cognitive differences between subjects in different areas? Showing the attitudes of all subjects alone was not enough to respond to the purpose of the study. In addition, the number of subjects in the two areas was very small. Whether they are representative is debatable. It is suggested that the author can supplement the relevant content.
- Consider that the article may be copied by others in the future. With regard to Figure 3-5, it is suggested that the author replace different colors with different patterns and add percentages. This could be friendly to the reader.
- In Section 4, it is recommended that authors can appropriately reduce the number of references to other people’s findings. Or, the findings of these others can be interpreted as hypotheses based on the results of the study.
- In Figure 2, the researchers show a lot of "numbers" and what do they mean? It is recommended that the author explain further.
- What are the possible reasons behind the results presented in Table 1? The author did not specify. It is recommended that the author supplement the relevant content.
- Finally, as a pilot study, we would like to know what the authors' follow-up plans are based on current research.
Reviewer 3 Report
My comments are as follows:
- The authors should add a new section about research theories related to the research model in Figure 2.
- The authors should investigate the measurement model's "composite reliability, validity of convergence, and validity on a discriminant scale" further.
- The authors should research more about hypotheses "Structural Model Evaluation & hypotheses testing" further. authors should add more discussion about measurement and structural modeling.
- What is the difference between this research and previous research?
Reviewer 4 Report
I think this this article could be improved by clearly sticking to the research question: capacity of students to-ward digital transformation in university teaching through three constructs: Motivations, Digital Pedagogy and Student Autonomy. This focus is not clear throughout the paper. I have annotated the paper. Including the table of how the three factors load would also be helpful as at least two items do not seem to load on the factor they are attributed to. The conclusions don't see pointed to the research but just conclusions in general about adjusting to the pandemic. Much room for improvement.

Round 2
Reviewer 1 Report
please see attached.

Author Response
|
Review 1 |
|
|
Major comments: |
|
|
1. The conclusions are supported by the results. |
We have modified the text. We have provided information on the results and implications. [477-499]. |
|
2. pdf |
The authors have carried out a review of the structural equation model. The authors have applied in the section “Data analysis” the procedure developed in the validation of the model. It can be verified that the items included in the factors have loads greater than .3 and that the model has good indices of goodness of fit according to the studies by Hair et al. (2014), Hoyle (1995), Hu & Bentler (1999) and Schumacker and Lomax (2016). |

Reviewer 2 Report
MDPI_ijerph-1506310-peer-review-v2 Review 1 Comments
The question raised last time was answered more adequately by the researchers. But there are the following questions, which I hope the author can improve. It can be published in principle after being supplemented. As mentioned last time, it is clear that COVID-19 will not end in the short term. There is still a lot of room for improvement in the online teaching model, and we also look forward to the follow-up research of researchers.
- 483 students who were registered in social and health sciences degrees in the academic year 2019-2020. It is suggested that authors can explain why they choose participants from these 2 disciplines? Is it because of objective conditions or other reasons?
- Regarding “10 Items”, the authors have re-adjusted and explained. However, it seems that there is something wrong with the logical relationship, or it may be because the author failed to discover it during the revision process. For example:
In line 367-369, the authors said: “Thus, initially 27 items were considered in 3 constructs for the delimitation of the model (see section: Data analysis). However, the items in the 3 categories were organized in the model confirmation process as follows:”
However, the author listed only 10 projects (page 6).
But, in line 425-427, the authors also said: “Of the 17 items that were integrated into the three factors, finally 10 of these were included in the confirmed model, since a satisfactory adjustment solution was obtained.”
It is suggested that the author could briefly explain the basic situation of the 27 items, and then state the reasons for the final reduction to 10 items, which is more conducive to the reader’s understanding.
- About Fig. 3-5, Since other readers or research may copy this article, the color graphics will become black and white, which will cause difficulties in identification. If you can, consider using different patterns. At the same time, the labeling of the numbers is best adjusted, and some numbers are too close to each other, which is easy to cause misreading.
- It is suggested that the author can improve the typography before the next submission. The pictures and the associated text are placed together as much as possible, rather than separated on different pages. At the same time, it is recommended that the author can proofread the full text again. Some texts may have errors due to formatting problems.
- In line 59," remote emergency teaching (ERT)" seems should be "emergency remote teaching (ERT)."
Author Response
|
Review 2. |
|
|
1. 483 students who were registered in social and health sciences degrees in the academic year 2019-2020. It is suggested that authors can explain why they choose participants from these 2 disciplines? Is it because of objective conditions or other reasons? |
The people that make up the sample of this research are university students belonging to different departments of the Faculty of Educational Sciences of the University of Seville (this being a public university of a face-to-face nature). Within this Faculty you can study the degrees of Physical Activity and Sports Sciences, Early Childhood Education, Primary Education, Pedagogy, Double Degree in Primary Education and French Studies, Double Degree in German Language and Literature and in Primary Education, Double Degree in Physiotherapy and Sciences of Physical Activity and Sports, Double Degree in Early Childhood and Primary Education. In addition, between the departments located in the center and those that teach, there are a total of 50 departments. Among the quality policy and objectives that are set in the center, it is worth highlighting the provision of training in university degrees aimed at excellence, internationalization and quality improvement to facilitate the employability of graduates. In this way, the questionnaire was sent to different departments of the Faculty. Exclusion criteria were not used. |
|
2. Regarding “10 Items”, the authors have re-adjusted and explained. However, it seems that there is something wrong with the logical relationship, or it may be because the author failed to discover it during the revision process. For example: In line 367-369, the authors said: “Thus, initially 27 items were considered in 3 constructs for the delimitation of the model (see section: Data analysis). However, the items in the 3 categories were organized in the model confirmation process as follows:”
However, the author listed only 10 projects (page 6).
But, in line 425-427, the authors also said: “Of the 17 items that were integrated into the three factors, finally 10 of these were included in the confirmed model, since a satisfactory adjustment solution was obtained.”
It is suggested that the author could briefly explain the basic situation of the 27 items, and then state the reasons for the final reduction to 10 items, which is more conducive to the reader’s understanding. |
We have made the suggested modifications. The authors have carried out a review of the structural equation model. The authors have applied in the section “Data analysis” the procedure developed in the validation of the model. It can be verified that the items included in the factors have loads greater than .3 and that the model has good indices of goodness of fit according to the studies by Hair et al. (2014), Hoyle (1995), Hu & Bentler (1999) and Schumacker and Lomax (2016). |
|
3. About Fig. 3-5, Since other readers or research may copy this article, the color graphics will become black and white, which will cause difficulties in identification. If you can, consider using different patterns. At the same time, the labeling of the numbers is best adjusted, and some numbers are too close to each other, which is easy to cause misreading. |
We have made the suggested modifications. |
|
4. It is suggested that the author can improve the typography before the next submission. The pictures and the associated text are placed together as much as possible, rather than separated on different pages. At the same time, it is recommended that the author can proofread the full text again. Some texts may have errors due to formatting problems. |
We have made the suggested modifications. |
|
5. In line 59," remote emergency teaching (ERT)" seems should be "emergency remote teaching (ERT)." |
We have made the suggested modifications |

Reviewer 3 Report
As a result, now that the paper is better than before, I propose the following comments:
- In the discussion and conclusions sections, the authors should highlight the contributions clearly and "explain the difference between this research and previous research."
Good luck
Author Response
|
Review 3
|
|
|
1. In the discussion and conclusions sections, the authors should highlight the contributions clearly and "explain the difference between this research and previous research."
|
The section has been included according to the reviewer's suggestion. |
